# Understanding Temporal Relations in Mandarin Chinese: An ERP Investigation

**DOI:** 10.3390/brainsci12040474

**Published:** 2022-04-03

**Authors:** Lijuan Chen, Yiyi Lu, Xiaodong Xu

**Affiliations:** 1School of Foreign Studies, Nanjing University of Posts and Telecommunications, Nanjing 210023, China; 20210009@njupt.edu.cn; 2School of Foreign Languages and Cultures, Nanjing Normal University, Nanjing 210097, China; endluyiyi@163.com

**Keywords:** temporal connectives, *before*, *after*, world knowledge, ERPs

## Abstract

Temporal connectives play a crucial role in marking the sequence of events during language comprehension. Although existing studies have shown that sentence comprehension can be modulated by temporal connectives, they have mainly focused on languages with grammatical tense such as English. It thus remains unclear how temporal information is processed in tenseless languages. The present study used event-related potentials (ERPs) to examine how world knowledge is retrieved and integrated in sentences linked by *zhiqian* (before) and *zhihou* (after) in Mandarin Chinese (e.g., *After/Before going to the countryside, Grandpa went to the city because the air there was fresh and pure*). The critical words (e.g., fresh) were either congruent or incongruent with world knowledge. Relative to the *after*-congruent sentences, the *after*-incongruent sentences evoked a P600 on critical words and a negativity on sentence-final words, whereas relative to *before*-congruent sentences, *before*-incongruent sentences showed no significant difference on critical words but a sustained negativity on sentence-final words. Additionally, *before*-congruent sentences elicited a larger sustained positivity (P600) than *after*-congruent sentences. The results suggest that *before* is more difficult to process than *after* in Mandarin Chinese, supporting the iconicity account of temporal relations.

## 1. Introduction

Time is an unbounded dimension. To explore time, people usually rely on a reference point or an anchor for location such as space, albeit abstractly [1,2,3]. Languages often employ two devices to locate situations in time: the grammatical device *tense* and the lexical device *temporal connectives* [4]. Temporal connectives play a pivotal role in marking temporal location, particularly in tenseless languages which lack overt inflectional marking of tense. Mandarin Chinese, for instance, is a tenseless language which does not mark past, present, or future with dedicated morphemes, yet Mandarin speakers successfully comprehend temporal information, largely depending on temporal adverbials (e.g., yesterday, last week), viewpoint aspect morphemes (e.g., ‘*le*’ termination or completion, ‘*zài*’ ongoing state), and temporal connectives (e.g., ‘*zhiqian*’ before, ‘*zhihou*’ after [5,6,7]). In Mandarin, the temporal connectives *zhiqian* (before) and *zhihou* (after) involve the selection of an event over a period of time, which provides important cues to set a time frame [8]. These temporal connectives have been demonstrated to be more difficult to understand than other connectives (e.g., causal connective *because*) [9], yet it remains unclear whether these temporal connectives are processed differently. Although a number of studies have investigated how sentence comprehension is modulated by temporal connectives such as *before* and *after* [10,11,12,13,14,15,16], they have mainly focused on tense languages such as English. It thus remains unclear how temporal connectives are processed in tenseless languages such as Mandarin Chinese.

### 1.1. The Comprehension of Before and After

The temporal connectives *before* and *after* are different in expressing iconicity. Linguistically, iconicity is the conceived similarity between the linguistic form and the meaning in the real world [17]. In other words, if the sequencing of clauses described in a sentence is consistent with the sequencing of events occurring in the real world, it is viewed as iconic; otherwise, it is viewed as non-iconic. Suppose *p* occurs before *q*: this temporal sequence can be described by an *after* construction [*After p, q*] in which the temporal order of the event is consistent with the chronological order, and therefore is regarded as iconic. The same event sequence, however, can also be described by a *before* construction [*Before q, p*], in which the sequence of the described clauses is inconsistent with the sequence of occurrence in the real world. It is therefore considered to be non-iconic. Similarly, in a construction in which the temporal connective occurs in the sentence-middle position, [*p before q*] is iconic whereas [*q after p*] is non-iconic (see Table 1). Previous studies adopting various approaches have shown that a non-iconic sequence is more difficult to understand than an iconic sequence. For example, Mandler [11] found that participants spent more time reading sentences when the order of occurrence mismatched the order in which they were presented. Münte et al. [12], using the event-related potentials (ERPs) technique, observed a larger sustained negative ERP modulation for non-chronological *before*-sentences (1a) than chronological *after*-sentences (1b) over the left frontal scalp. Interestingly, the ERP difference was significantly correlated with a working memory score. Individuals with higher working memory span showed a more pronounced negative difference between *before* and *after* sentences, suggesting that computing a non-iconic temporal relation places high demands on the working memory system. Ye et al. [16] found that the caudate nucleus and middle frontal gyrus were more strongly activated for processing *before*-clauses than *after*-clauses. Moreover, another line of studies has revealed that sentence-initial sentences headed by *before* are more difficult to understand than those headed by *after* for young children [18] as well as Parkinson patients [19,20].

(1a)*Before* the scientist submitted the article, the journal changed its policy.(1b)*After* the scientist submitted the article, the journal changed its policy.

The comprehension of *before* and *after* could also be influenced by their position in a sentence. Politzer-Ahles et al. [14] compared the neural correlates of processing *before* and *after* in both sentence-initial (e.g., 1a/b) and sentence-middle positions (e.g., 2a/b). They found that *before* elicited more negative ERP responses than *after* in the sentence-initial position but more positive ones in the sentence-middle position. The result in the sentence-initial context is consistent with previous studies (e.g., [12]), but the sentence-middle result shows the opposite, indicating that ERP modulations elicited by temporal connectives are related to the order in which events are presented.

(2a)The journal changed its policy *before* the scientist submitted the article.(2b)The journal changed its policy *after* the scientist submitted the article.

The studies addressed so far seem to support the idea that a non-iconic temporal relation is semantically/pragmatically more costly to understand than iconic temporal relation (regardless of connective type). However, despite the difference in expression iconicity, temporal connectives also differ in frequency and polarity, and the explanation based on frequency and polarity will have a different prediction for processing temporal connectives such as *before* and *after*.

According to the polarity-based explanation, *before* shares a similar meaning with *in front of* in the pair *in front of–behind*, in which *in front of* is positive as it is used to describe the visible area of perception, whereas its counterpart *behind* is negative as it describes the invisible area [21,22]. Linguists therefore distinguish them by labeling *before* as positive and *after* as negative. Indeed, psycholinguistic evidence has demonstrated that people tend to acquire positive words such as *before* more easily than negative words such as *after* [23]. In an early behavioral study, Clark [10] found that children had greater difficulty comprehending *after*-clauses than *before*-clauses.

With regard to frequency of use as temporal connectives, *after* is used less often than *before* in both adults and children’s language usage [24,25,26]. Besides functioning as a temporal connective, *after* is more frequently used as a preposition (like *behind*), and therefore has a less consistent mapping between form and meaning as compared to *before* in terms of temporal connective. This variability may lead to greater uncertainty in interpreting *after* sentences compared to *before* sentences, meaning that understanding *after* sentences could be cognitively more demanding.

Notably, there are cross-linguistic differences in expressing iconicity with temporal connectives. For example, the relationship between temporal connectives and iconicity in Mandarin Chinese is much clearer than in English, because in Mandarin, temporal connectives *zhiqian* (before) and *zhihou* (after) only occur at the non-initial position of a sentence. In this case, *zhiqian* generally expresses a non-chronological temporal relation whereas *zhihou* expresses a chronological temporal relation. This unambiguity provides a better opportunity to test the iconicity hypothesis.

In summary, regarding the different properties of *before* and *after* and especially their discrepancies between English and Chinese, it remains uncertain which of the two temporal connectives is cognitively more difficult to process. If iconicity is the most relevant factor for processing, then *before* should be more cognitively demanding to process than *after*. If, however, the value of polarity and form–meaning mapping (frequency) is more important, then *after* should be more cognitively demanding than *before*.

### 1.2. How World Knowledge Is Activated during Temporal Connectives Processing

During language comprehension, readers not only rely on semantic and syntactic knowledge, but also depend on world knowledge to establish a coherent sentence/discourse model [27,28,29,30,31,32]. If the linguistic input is inconsistent with world knowledge, a negative-going brain response, appearing around 300–500 ms (i.e., N400), will be regularly elicited [27,28,29,30,31]. However, whether a semantically reasonable sentence is pragmatically valid could be constrained by the coherent markers, namely connectives. A number of recent studies have explored this issue. Using event-related potentials (ERPs), Xu et al. [31] investigated how world knowledge incongruency was modulated by causal and concessive connectives (‘*yinwei*’ because vs. ‘*jinguan*’ although). They found that causal and concessive connectives had different influences on pragmatic processing at different stages of sentence processing. During early retrieval processing, world knowledge anomalies (e.g., Harbin is *warm* in winter) evoked indistinguishable N400 effects in both causal and concessive structures; at the integration stage, however, they showed dissociable ERP effects: while violation of world knowledge elicited a P600 effect in causal structures, a late negativity (N600) effect was obtained in concessive structures.

Only a few studies have explored how different temporal information influences world knowledge processing. Rinck et al. [15] presented seven-sentence passages to their participants. In each passage, the temporal information conveyed at target sentence (e.g., *Claudia was already waiting for him when he got off the train with his huge bag*) was either consistent (e.g., *Markus’s train arrived at Dresden Central Station 20 **min** after Claudia’s train*.) or inconsistent (e.g., *Markus’s train arrived at Dresden Central Station 20 **min** before Claudia’s train*.) with the contextual sentences. The results showed that inconsistent sentences were processed longer than consistent ones. In an ERP study, Xiang et al. [33] presented not only ad hoc stimuli with ambiguous veridicality (similar to 2a/b in Münte et al. [12], but also real-world stimuli (e.g., *Before/After Tiger Woods won the Masters, Jake started playing golf*) which are veridical based on world knowledge. The result showed that *before*-sentences evoked a larger sustained negativity than *after*-sentences in ad hoc stimuli, but no effect was observed in real-event stimuli, suggesting that the difference underlying processing of these two types of temporal relations could be related to the uncertainty of the events. Nieuwland [13], using ERP techniques, also investigated how world knowledge was processed in *before* vs. *after* sentences.

(3a)After the global economic crisis, securing a mortgage was harder.(3b)After the global economic crisis, securing a mortgage was easy.(3c)Before the global economic crisis, securing a mortgage was easy.(3d)Before the global economic crisis, securing a mortgage was harder.

The results demonstrated that while the world knowledge-inconsistent words (3b) elicited a larger N400 than the world knowledge-consistent words (3a) in *after* sentences, the N400 difference effect was largely attenuated in *before* sentences (3c/d). However, at the late time window, the inconsistent words (3d) elicited a larger P600 than the consistent words (3c) in *before* sentences. These results indicate that integrating mismatched pragmatic information into discourse could be delayed in a non-chronological temporal sequence in comparison to a chronological sequence.

### 1.3. The Present Study

In the current study, we aimed to investigate how world knowledge information was retrieved and integrated into clauses which were linked by temporal connectives *zhiqian* (before) and *zhihou* (after) in Mandarin Chinese.

Different from English in which temporal information (e.g., past, present, and future) can be expressed by inflectional morphology, temporal information in Chinese is expressed by lexical means such as time adverbials (e.g., yesterday), viewpoint aspect morphemes (e.g., *le, zai*), and discourse contexts (connectives, *before, after*). The lack of a tense marking system will influence the perception of temporal sequence in Chinese. In the framework proposed by Reichenbach [34], temporal information in language involves three time points: speech time (the time of the utterance), reference time (the time of perception by the listener), and event time (the time when the event occurs). While processing temporal information, it is necessary to grasp the relationship between speech time and reference time, which is linguistically encoded by tense. In a language which lacks tense marking such as Mandarin, where the relationship between speech time and reference time is usually ambiguous, it tends to be more difficult to comprehend reference time misalignment (RTM). Li et al. [35], for example, found that RTM between subordinate clauses and main clauses in English could be covertly detected by native English speakers even when they were not overtly noticed, as evidenced by an N400 effect, but even highly proficient Chinese–English bilinguals could not detect RTM, and showed no N400 effect.

Although *zhiqian* and *zhihou* share the property of their English counterparts *before* and *after* in expressing temporal information, they differ in a number of aspects, especially for *zhiqian*. *Zhiqian* in its current use still retains ancient characteristics, which is different from *before* in English. *Zhi* in ancient Chinese can be interpreted as the structural auxiliary *de* as in modern Chinese, meaning *in front of* when it is merged with *qian*. *Zhiqian* in its current use has maintained some of the ancient Chinese meaning, which is quite different from *before* in English. *Zhiqian* shares the properties of *zhi* in ancient usage and also retains the possessive property in the temporal proximity, and *zhiqian* provides the time reference to locate the target event in the recent past, which is close to the speech time [36]. *Zhihou* has developed maturely in modern Chinese, losing the ancient Chinese boundary proximity meaning of *zhi* to some degree, and *zhihou* is commonly regarded as a whole, and provides the time reference to locate the target event in the future, which may be far from the speech time.

As shown in Table 2, the experiment sentences have the structure “NP_human_ goes to place A + *before*/*after* + going to place B, because + (the typical feature) + there + [critical word] + [sentence-final word]”. As discussed in the previous section, [*p zhiqian q*] is non-iconic because in the sentence (4c), the event ‘going to the countryside’ happens before the event ‘going to the city’, and thus the order of real event is contrary to the order of mention. Similarly, [*p zhihou q*] is iconic in the sentence (4a) because the order of mention is consistent with the real-world event sequence. In addition, *le* is generally viewed as a perfective marker [37]. Therefore, in the construction “*p zhihou q*” (4a/b), the event sequence should be (*p*, *q*), and *p* is veridical as *le* in the clause *q* denotes that the events (both going to countryside and going to the city) actually happened. *q* entails *p*, thus *after*-sentences are veridical. In contrast, in the construction “*p zhiqian q*” (4c/d), the event sequence should be (*q*, *p*), and *q* does not entail *p* (Table 1). Therefore, whether the *before*-sentences are veridical or not remains ambiguous. Although the locative pronoun *nali* (there) could be interpreted as referring to a closer or more distant place (Noun), the clauses would be acceptable if *because* licenses a reason for the movement (e.g., Grandpa liked to go to a place because the air there was pure and fresh). In other words, world knowledge (e.g., the city is usually a place where air pollution is serious, whereas the countryside is normally a place with fresh air.) helps to causally connect the main clause with the subordinate clause. Mandarin speakers have a preference to interpret the locative pronoun as referring to the closer noun in both *after* and *before* sentences. According to this preference, sentences 4a and 4c are acceptable because the referential interpretation in these two conditions (i.e., countryside) is consistent with common sense that the air in the countryside is fresher and purer. As for sentences 4b or 4d, however, due to the two places being swapped, the closer referential interpretation of the locative pronoun (i.e., city) is inconsistent with readers’ world knowledge.

Thus, the experiment had a 2 (connective type: *before* vs. *after*) × 2 (world knowledge congruency: congruent vs. incongruent) factorial design with both connective type and world knowledge information being manipulated.

The aim of the present study is to test the neural correlates of processing different temporal connectives and their potential influence on truth value judgments. Based on previous research [12,13,14], we had two lines of predictions. Firstly, if the iconicity account holds in processing Chinese temporal relations as in previous studies (e.g., [12]), a chronological temporal relation should be easier to process than a non-chronological relation. Therefore, pragmatic anomalies in *zhihou* (after) sentences should be quickly detected and initiate a repair process to re-interpret the incongruency, which will possibly be manifested by an N400 or N400/P600 effect. As for *zhiqian* (before) sentences, however, due to the additional process of reversing a non-chronological temporal sequence, the comprehension system may not be rapidly sensitive to the anomalies until a later processing stage in which more constraints are available. Under such circumstances, the pragmatic anomalies may induce reduced or even null N400/P600 effects (see [13]). However, according to the polarity-based account and the form–meaning mapping account [24,25,26], if *after*-sentences are more difficult to process than *before*-sentences, as observed in previous studies (e.g., [9,26]), then a reverse pattern of ERP modulations should be expected; that is, the above-mentioned ERP modulations should be obtained in *before* sentences instead of *after* sentences. In particular, given the important role of working memory in establishing a non-chronological temporal representation (Münte et al. [12]), we might find a significant correlation between the ERP modulations obtained at the connectives and readers’ working memory score. Finally, in addition to the critical words, the ERP effects underlying processing different temporal connectives should also be manifested at the connectives [12,14].

## 2. Materials and Methods

### 2.1. Participants

Thirty-two undergraduate or postgraduate native speakers of Mandarin (13 male, mean age = 21.8, SD = 2.92, range 18–26) took part in the ERP experiment. Four participants were excluded from data analysis because of excessive EEG artifacts. All participants were right-handed and had normal or corrected-to-normal vision and none of them had a history of neurological or psychiatric impairments. Each participant gave informed consent before the formal experiment and was compensated for participation.

### 2.2. Design and Materials

One hundred and twenty sets of two-clause sentences were developed (see Table 2 for example stimuli). The main clause described an event in which someone moved to one place *before* or *after* moving to another. The subordinate clauses were kept the same across conditions, so the differences occurred only in the main clause where the type of connective and the mention order of the two places were manipulated. Since the locative pronoun is preferentially interpreted as referring to the closer referent, such co-referential relation would cause the critical word to be congruent (the air in the countryside was fresh) or incongruent (the air in the city was fresh) with one’s world knowledge in *after* sentences (4a/b) or *before* sentences (4c/d).

Before the EEG recording, the selected set of sentences underwent four pretests: a cloze probability test, a sentence rating test and two forced-choice tests. One hundred fifty quadruplets of sentences were developed for the cloze probability test, and 140 quadruplets were left for the other three pretests after removing 10 quadruplets with lower cloze probability scores.

In the cloze probability test, 36 participants were randomly assigned to one of four counterbalanced lists with the last two words truncated (e.g., *After going to the city, Grandpa went to the countryside because the air there was …*) and were asked to record the first word that came to their mind and to make the completed sentence as natural as possible. Cloze value was calculated as the percentage of the participants who completed the sentences with the intended critical words [38]. Mean cloze probability scores per condition are shown in Table 3. The results revealed a significant main effect of congruency (*F*(1, 149) = 358.5, *p* < 0.001) without any interaction or other main effect, demonstrating that the critical words were equally predictable in *after* and *before* sentences. In the sentence acceptability test, another group of 31 students were assigned to one of four counterbalanced lists in which they were asked to rate the overall acceptability of the sentences on a 7-point Likert Scale (1 = highly unacceptable, 7 = highly acceptable). Mean scores and standard deviations are given in Table 3. Statistical analysis revealed a significant main effect of congruency (*F*(1140) = 1557.62, *p* < 0.001) and a connective type × congruency interaction (*F*(1140) = 5.51, *p* < 0.05). Follow-up tests showed that the *after*-congruent sentences were more acceptable than *before*-congruent sentences (*t*(140) = 3.05, *p* < 0.01). As *nali* is a syntactically unmarked demonstrative pronoun in Chinese, it can refer to either the closer (place B) or the more distant place (place A) in a sentence, depending on the contextual preference [31,39,40]. To this end, two forced choice tests were conducted. The first test presented the complete sentences and 30 participants were then asked to judge whether *nali* refers to place A, place B or a third place unmentioned in the context. In the second forced choice test, the sentences were truncated after the demonstrative pronoun *nali* (e.g., *after going to the city, Grandpa went to the countryside because there …*); another group of 37 participants were asked to select the actual referent of *nali* among the three potential choices, as carried out in the first forced choice test. As shown in Table 4, the locative pronoun *nali* was preferentially interpreted as referring to the closer location (place B) throughout the sentences. In the *before* sentences, the closer referential preference was stronger in the congruent than in the incongruent condition (*t*(140) = 4.86, *p* < 0.001), whereas in the *after* sentences, the closer referential bias was equally strong irrespective of the sentence congruency (85.7% vs. 83.2%, *p* > 0.5). Interestingly, in the congruent conditions, the closer preference was stronger in the *before* sentences than in the *after* sentences (*t*(140) = 3.12, *p* < 0.01) when the truncated information was added. On the contrary, in the incongruent conditions, the distant referents became more preferential than the closer ones.

In the ERP study, the 120 experimental items selected based on the pretests (60 for each temporal connective type) were distributed across 4 different testing lists according to a Latin square procedure, with each list containing 30 items per condition. In addition, 120 two-clause filler items were interspersed randomly among the 120 experimental items to obscure the purpose of the experiment from participants. The 120 filler items were always consistent with real world knowledge, including 40 two-clause sentences connected by *before* or *after*; 40 sentences in which the two clauses were connected by other connectives (e.g., *although, therefore*), and 40 sentences without connectives. The experimental items were pseudo-randomly mixed with fillers to limit the succession of no more than three sentences of identical type and no more than three sentences had comprehension questions with the identical intended answer (e.g., *Does Grandpa like the fresh air?* and the participant press the YES button). Participants were randomly assigned to read one of the testing lists.

### 2.3. Procedures

Participants were seated comfortably in a chair in a dimly lit and sound-attenuated room where stimuli were presented on a computer screen. Each trial started with a fixation cross (“+”) at the center of the screen for 500 ms. After a 500 ms blank screen, each sentence was presented word-by-word with a single word duration of 400 ms followed by a 400 ms blank screen. Participants were then required to answer a yes/no comprehension question by pressing one of two keys as quickly and accurately as possible. The questions were designed to probe participants’ understanding of the sentences. About 60% of the trials were followed by a comprehension question. For the remaining trials, the word “blank” was presented following the trials and participants had to press the space bar to continue. The assignment of left/right hand to yes/no response was counterbalanced across participants. Each participant performed 20 practice trials at the beginning of the experiment, which were similar in structure to the test stimuli. The stimuli were divided into four blocks, and the participant had a break of about 5 min between each block. The whole experiment lasted about 2 h per participant, including electrode preparation, instructions, and practice. At the end of the ERP experiment, the participants were asked to complete a working memory test (reading span). In the working memory test, participants pressed a key to judge whether the sentence presented on the screen was true or not based on their common sense and, at the same time, memorized the last word of each sentence. After a series of sentences (the number of sentences gradually increasing), participants needed to type in the memorized words in order. Reading span capacity scores were calculated by summing the number of words recalled in correct sequence with correct judgment of each sentence [41,42].

### 2.4. EEG Recording and Analysis

The EEG response was recorded from 31 electrodes held in place by an elastic cap (Electro-cap International). Additional electrodes were placed above the right eye and at the outer canthus of the left eye to monitor the vertical and horizontal electrooculogram. The EEG was referenced online to the tip of the nose and re-referenced offline to the algebraic average activity measured in the left and right mastoids (TP9 and TP10). Electrode impedance was kept below 5 kΩ. EEG signals were filtered using a band-pass of 0.016–70 Hz and digitized at a sampling rate of 500 Hz to exclude slow drifts and then was segmented into epochs from −200 to 900 ms around the critical words (e.g., *fresh*) or the sentence-final words (e.g., *pure*), with a prestimulus baseline of 200 ms. The signal was averaged for each condition and each participant before grand averages were computed across all participants. An independent component analysis (ICA) was applied to correct ocular artifacts. After correction, remaining artifacts (EOG, movement, technical) were marked manually and dropped from data analysis.

Based on the relevant literature in sentence/discourse processing [39,43,44], statistical analyses of the ERPs were carried out in the time window of 300–800 ms. ANOVAs were conducted on mean ERP amplitudes in this time window with temporal connectives (*before* vs. *after*), congruency (*congruent* vs. *incongruent*) and topographical factors as within-participant variables. For the midline analysis, the topographic factor was electrode (three levels): anterior (Fz), central (Cz), and posterior (Pz). For the lateral analysis, the topographic factor was region (three levels: anterior vs. central vs. posterior) and hemisphere (two levels: left vs. right). Thus, six regions of interest (ROIs) were calculated from the mean value of three electrodes each: anterior left (AL: F3, F7, FC5), central left (CL: FC1, CP1, CP5), posterior left (PL: P3, P7, O1), anterior right (AR: F4, F8, FC6), central right (CR: FC2, CP2, CP6), and posterior right (PR: P4, P8, O2) (Kulakova et al., [45]). Mean amplitudes over electrodes in each region of interest were entered into ANOVAs. The Greenhouse–Geisser correction was applied when appropriate [46]. In the correlation analysis on connectives, we chose the 3000–4500 ms time window, which, according to a previous relevant study [12], showed the most pronounced effect for temporal connectives.

## 3. Results

### 3.1. Behavioral Results

The mean accuracy in the comprehension task was 93.9%, suggesting that the participants could understand the sentences well. The average accuracy was 95.4% for *after*-congruent sentences and 92.2% for *after*-incongruent sentences, whereas the accuracy was 93.8% for *before*-congruent sentences and 94.0% for *before*-incongruent sentences. A 2 × 2 repeated measures ANOVA showed a marginally significant main effect of congruency (*p* = 0.054), indicating that incongruent sentences were more difficult to comprehend than congruent sentences.

### 3.2. ERP Results

The grand-average ERPs, time-locked to the critical words and the sentence-final words, are shown in Figure 1 and Figure 2, respectively. As can be seen in Figure 1, *after*-incongruent sentences elicited a sustained larger positivity (P600) relative to the *after*-congruent sentences, whereas there was no difference between the incongruent and congruent conditions in *before* sentences. Moreover, *before*-congruent sentences elicited a larger positivity than *after*-congruent sentences. At the sentence-final position, incongruent sentences elicited greater negativities than congruent sentences irrespective of the connective types, as demonstrated in Figure 2.

#### 3.2.1. ERP Responses to the Critical Words (300–800 ms Time Window/P600)

In ERP data analysis, we carried out two omnibus repeated-measures ANOVAs in which the mean amplitude (ROI-based) from the lateral electrodes and the midline electrodes were analyzed separately. This approach has been adopted by a large number of ERP studies including those focusing on concessive relations [31,47]. Below, we mainly report the statistically significant results. For the full ANOVA results, please refer to the Appendix A (Table A1
*presents the results for the critical words and*
Table A2
*the results for the sentence-final words*).

In the lateral analysis, there was only a significant interaction between *connective type* and *congruency* (*F*(1, 27) = 9.40, *p* < 0.01, η^2^ = 0.258). Follow-up tests showed the *after*-incongruent sentences evoked more positive ERPs relative to the *after*-congruent sentences (*t*(1, 27) = 5.03, *p* < 0.01), whereas there was no significant difference between *before*-congruent and *before*-incongruent sentences (see Figure 3, the left panel). Moreover, *before*-congruent sentences elicited larger positive ERPs than *after*-congruent sentences (*t*(1, 27) = 2.93, *p* < 0.01). Similarly, the midline analysis also revealed a robust *connective type* × *congruency* interaction (*F*(1, 27) = 5.07, *p* = 0.033, η^2^ = 0.158). Follow-up tests revealed that *after*-incongruent sentences evoked more positive ERPs than the *after*-congruent sentences (*t*(1, 27) = 2.85, *p* < 0.01) whereas there was no significant difference between *before*-incongruent and *before*-congruent sentences (see Figure 3, the right panel). There was only a marginally significant difference between the *before*-congruent and *after*-congruent sentences (*F*(1, 27) = 3.200, *p* = 0.085, η2 = 0.106). No other interesting main effects or interactions were significant.

#### 3.2.2. ERP Responses to the Sentence-Final Word (300-800 ms Time Window/Negativity)

At lateral sites, ANOVA revealed a significant main effect of *congruency* (*F* (1, 27) = 6.01, *p* = 0.021, η^2^ = 0.182) and an interaction between *congruency* and *hemisphere* (*F* (1, 27) = 5.00, *p* = 0.034, η^2^ = 0.156). No other interesting main effects or interactions were observed. The follow-up analysis to resolve the interaction showed that incongruent sentences elicited larger negativities than congruent sentences in the right hemisphere (*t* (1, 27) = 3.24, *p* < 0.01) but not in the left. No significant effects were found in the midline analyses.

#### 3.2.3. Correlation Analyses

Following the previous study [12], we calculated the ERP difference effects at the connectives and carried out correlation analysis between the ERP difference effects and the individuals’ working memory span (see Figure 4). The difference effect at the connectives was the amplitude difference between *before* sentences (averaged over all *before* sentences) and *after* sentences (averaged over all *after* sentences). In the time window of 3000–4500 ms, the ERP difference amplitude (*before* vs. *after*) in the left posterior area (P3, P7, O1) was positively correlated with individuals’ working memory span (*r*(28) = 0.441, *p* = 0.019).

## 4. Discussion

This study investigated how world knowledge is modulated by the temporal connectives *before* and *after* in Mandarin Chinese. Behavioral results showed that comprehension accuracy for incongruent sentences was lower than those for the congruent sentences regardless of the connective type. As for the ERP findings, critical words in *after*-incongruent sentences evoked a larger positivity than critical words in *after*-congruent sentences, whereas the critical words in *before*-incongruent and *before*-congruent sentences revealed no significant difference. For congruent sentences, critical words in *before*-congruent sentences elicited larger positivity than critical words in *after*-congruent sentences (see Figure 1). Moreover, at the connectives, the ERP difference between *before* sentences and *after* sentences was positively correlated with individuals’ working memory (see Figure 4). As for sentence-final words, incongruent sentences elicited larger negativities than congruent sentences irrespective of temporal relations (see Figure 2).

### 4.1. Effect of Temporal Connectives on Pragmatic Processing at the Critical Words

Pragmatic incongruency has different ERP modulations on *before* vs. *after* sentences. *After* sentences with pragmatic anomalies evoked a P600 effect, but *before* sentences did not. In addition, we also observed a significant positive correlation between the ERP difference at the connectives (*before* minus *after*) and individuals’ working memory span, consistent with Münte et al. [12] and Xiang et al. [33], suggesting that participants with higher working memory capacity are better at dealing with the reverse temporal relations. This seems to support the iconicity account, namely that comprehending *before* temporal clauses is more difficult than understanding *after* temporal clauses.

While comprehending temporal relations, people tend to map the linear order of the temporal clause to the chronological order of the events in the real world. Since the event described in *after* sentences is consistent with such a preference, readers have little difficulty processing this temporal relation. They, therefore, can successfully detect the pragmatic anomalies and initiate a reinterpretation process to rationalize the inappropriate input, as reflected by a P600 effect [48,49]. As for *before* sentences, however, readers need to make more effort to build a new mental representation because the *before* construction not only reverses the linear order but also requires readers to integrate the context with world knowledge to verify its validity [40,50]. Since this process can be cognitively demanding, fewer cognitive resources are available for processing pragmatic anomalies, resulting in two indistinguishable positivities. Similar “insensitivity to semantic/pragmatic anomalies” have been repeatedly observed in studies in which complex sentence/discourse structures with ambiguous representations were adopted (e.g., *reversed* temporal relation, Nieuwland 2015; *concessive* relation [31,40]; *counterfactual* relation, [51,52]). Of particular relevance, Nieuwland [13] found that while pragmatic anomalous words elicited a greater N400 effect in *after* sentences, this difference effect was significantly attenuated in *before* sentences. Nonetheless, the absence of a P600 effect on the critical words in the current study (in the *before* construction) does not mean that the pragmatic incongruence exerts no influence on subsequent sentence processing. Indeed, it modulates the late stage of integrating sentence constituents into the discourse representation, as manifested by an increased negativity for the incongruent relative to the congruent sentences at the wrap-up position. We will return to this issue in the last section.

The different cognitive processes underpinning *after* and *before* sentences can also be demonstrated by directly comparing the two consistent conditions. The observation of a larger sustained positivity (P600) for *before*-congruent sentences than for *after*-congruent sentences at critical words indicates that truth value judgments under a reversed temporal relation are more difficult than a sequential temporal relation. However, although previous research on sentence processing has recorded a close relation between P600 effect and sentence-level prediction (e.g., the predictability of an upcoming word) and sentence acceptability (e.g., [53]), the larger P600 modulation for *before* sentences cannot simply be attributed to such effects because critical words are equally predictable and the sentences are equally acceptable between the two constructions (cloze probability: 48.7% vs. 49.7%; acceptability: 5.35 vs. 5.13). Instead, the P600 modulation is more likely related to the complex semantic and pragmatic properties denoted by the reverse temporal expression *before* compared to *after*. This is also consistent with Nieuwland [13], who observed a negative ERP effect (N400) in response to pragmatic inconsistencies in *after* sentences but not *before* sentences. However, one difference between the present study and Nieuwland’s [13] study is that we found a P600 but they reported an N400. This could be related to the linguistic differences of expressing temporal relations between Chinese and English. As a tenseless language, time arguments in Chinese are regularly absent. Chinese speakers depend on a number of approaches, such as deictic patterns (e.g., next week) and context cues, to express temporality [6]. As a consequence, the truth value judgement in a temporal structure in Chinese is not simply based on the lexical semantics of a sentence, but rather depends on the explicit temporal devices, the contextual cues, and even readers’ personal social cognition capacities [6,54,55,56]. However, different from the present study, Nieuwland also reported a late positivity effect at the critical words in *before* sentences, which was interpreted as reflecting the involvement of monitoring processes [13]. These differences may be related to the recruitment of different computing processes. Firstly, the sentence position of the connectives was different between the studies, which can influence the ease of retrieving and keeping information in working memory, and may consequently influence the process of building a discourse representation. When a connective appears at the beginning of a sentence (e.g., [13]), it can provide immediate information about the temporal relationship of the statement at an early stage; thus, encountering pragmatic anomalies can immediately evoke an N400 in *after* sentences or a late P600 in *before* sentences. When a connective appears in the sentence-middle position, however, due to its closer distance to subsequent critical words, repairing and integrating the incongruency into discourse could increase processing load and, therefore, prolong processing time. A similar prolonged reinterpretation process, as reflected by a sustained negativity in post-critical words, has been reported for other complex constructions containing reverse coherence connectives (e.g., *even so* construction in English, Xiang and Kuperberg [47]; *lian … dou* (even) construction in Chinese, Jiang et al. [29]). Secondly, different from Nieuwland [13] and Politzer-Ahles et al.’s [14] studies, the sentences in the current study are more complex. A co-referential relation must be established between the two clauses, as there is a locative pronoun (*nali*) in the subordinate clause which must be resolved towards one of the locations in the preceding clause. The co-referential relation is more difficult to establish in the reversed temporal relation than in the canonical temporal relation. This could also contribute to the delayed negative-going effect.

### 4.2. Pragmatic Incongruence Effect in Sentence-Final Position

The processing cost of violating a sentence’s truth value may not fully manifest on the critical word, but instead be delayed to a later stage at which multiple constraints are integrated into the discourse representation. Indeed, pragmatic incongruencies evoked larger negativities at the sentence-final position in both *after* and *before* constructions, even though only the *after* construction evoked a P600 effect at the critical word. However, although the pragmatic incongruence evoked a late negativity in both constructions, the underlying cognitive processes as well as the functional interpretation should not be the same. For *after* sentences, while the P600 reflects a conflict between language input and knowledge stored in long-term memory and initial attempts to repair such incongruency, the late negativity at the sentence-final position may reflect a prolonged second-pass process to reinterpret the incongruency before integrating it into discourse. A similar pattern has been reported by Xiang and Kuperberg [47] who found that pragmatic inconsistencies evoked a large, late negativity at the sentence-final position in addition to early modulations at the target, irrespective of whether the sentence was connected by a complex concessive connective (i.e., *even so*) or not. As for *before* sentences, pragmatic anomalies only elicited a late negativity at the sentence-final position. Nevertheless, different from the negativity in *after* sentences, the negativity in *before* sentences lasted longer and was distributed more extensively over the scalp, which is comparable to the sustained negativities obtained for complex event structures where a new event representation has to be updated (e.g., [29,31,57,58,59]. Instead of simply initiating a repair process when encountering a non-iconic temporal relation, the comprehension system must first reorder a reverse temporal sequence and then reinterpret the pragmatic incongruency. Such a process could be cognitively demanding for the system and therefore delayed to the sentence wrap-up position.

## 5. Conclusions

The present ERP study examined how world knowledge is influenced by the temporal connectives *before* and *after* in Mandarin Chinese, a language without tense. It was found that at the critical word, *after*-incongruent sentences evoked a sustained late positivity relative to the *after*-congruent sentences, whereas *before*-incongruent sentences and *before*-congruent sentences evoked indistinguishable ERP responses. An increased sustained negativity was elicited by sentence-final incongruent words compared to congruent ones irrespective of temporal relations. These results suggest that *before* is more difficult to comprehend than *after* in Mandarin, supporting an explanation of processing temporal sequence in terms of iconicity. In the future, it would be interesting to compare how temporal relations are processed in Chinese and other related languages (e.g., Korean, Japanese).

## Figures and Tables

**Figure 1 brainsci-12-00474-f001:**
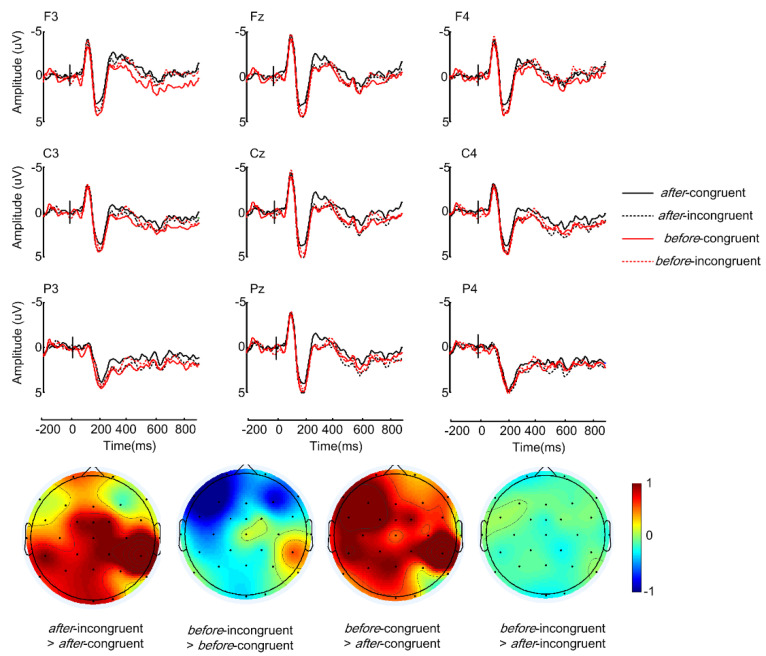
ERP results to the critical words in each of the four conditions at 9 representative electrodes are shown in the graph. Scalp distributions of the relevant difference effects (in the 300–800 ms time window) are given below the ERP graph.

**Figure 2 brainsci-12-00474-f002:**
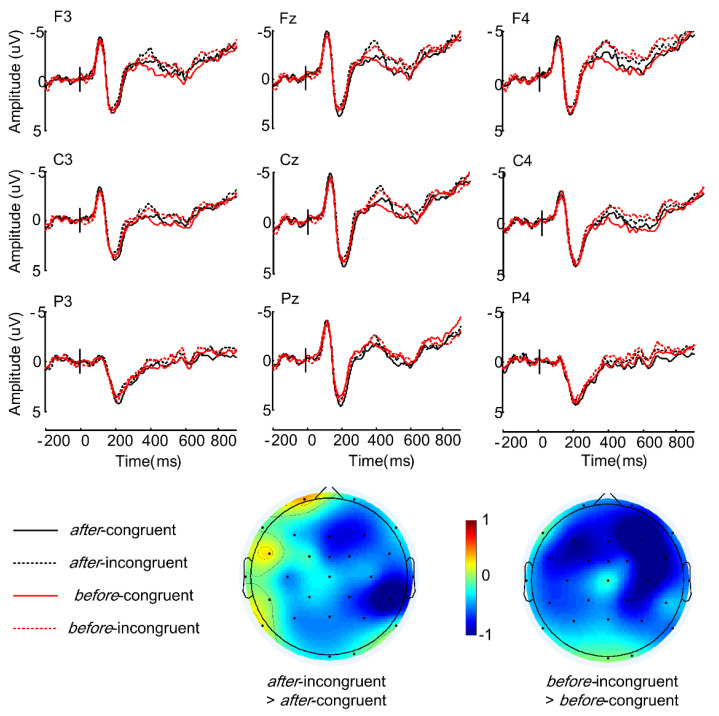
ERP responses to sentence-final words in each of the four conditions at 9 representative electrodes are shown in the graph. Scalp distributions of the relevant difference effects (in the 300–800 ms time window) are given below the ERP graph.

**Figure 3 brainsci-12-00474-f003:**
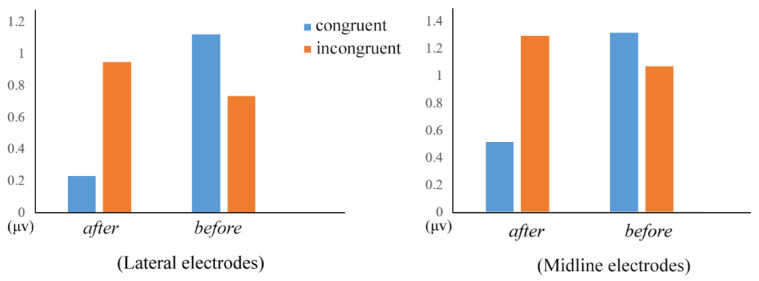
The bar plots for the average P600 amplitude (300–800 ms) of the critical words in each experimental condition in lateral analysis (**left**) and midline analysis (**right**).

**Figure 4 brainsci-12-00474-f004:**
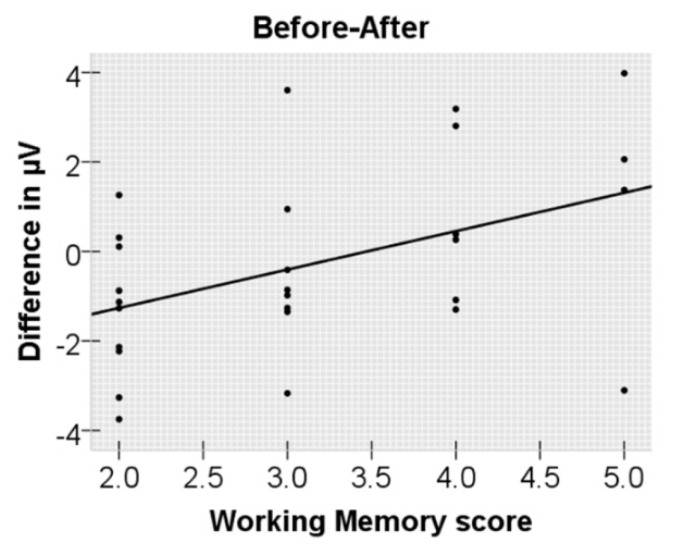
Correlation of working memory score and ERP voltage difference (at the connectives, 3000–4500 ms) between *before* and *after* sentences over the left posterior electrodes.

**Table 1 brainsci-12-00474-t001:** *Before* and *After* in semantic accounts.

	Semantic Accounts		
	Veridicality	Iconicity	Polarity
p *before* q	non-veridical	iconic	positive
q *after* p	veridical	non-iconic	negative
*Before* q, p.	non-veridical	non-iconic	positive
*After* p, q.	veridical	iconic	negative
q *zhiqian* p	non-veridical	non-iconic	positive
p *zhihou* q	veridical	iconic	negative

Note: *Veridicality* refers to the degree to which an experience, perception, or interpretation accurately represents reality.

**Table 2 brainsci-12-00474-t002:** Experimental conditions and example sentences with literal translations and glosses.

Conditions	Examples
*after*- *congruent*	(4a) 爷爷/去/城里/**之后**/去了/乡下，因为/那里/空气/清新/如洗。 Yeye/qu/chengli/**zhihou**/qule/xiangxia/,/yinwei/nali/kongqi/qingxin/ruxi. Grandpa/go-to/city/**after**/went/countryside, because/there/air/fresh/as washing. *After going to the city, Grandpa went to the countryside because the air there was fresh and pure.*
*after*- *incongruent*	(4b) 爷爷/去/乡下/**之后**/去了/城里，因为/那里/空气/清新/如洗。 Yeye/qu/xiangxia/**zhihou**/qule/chengli/,/yinwei/nali/kongqi/qingxin/ruxi. Grandpa/go-to/countryside/**after**/went/city, because/there/air/fresh/as washing. *After going to the countryside, Grandpa went to the city because the air there was fresh and pure.*
*before*- *congruent*	(4c) 爷爷/去/城里/**之前**/去了/乡下，因为/那里/空气/清新/如洗。 Yeye/qu/chengli/**zhiqian**/qule/xiangxia/,/yinwei/nali/kongqi/qingxin/ruxi. Grandpa/go-to/city/**before**/went/countryside, because/there/air/fresh/as washing. *Before going to the city, Grandpa went to the countryside because the air there was fresh and pure.*
*before*- *incongruent*	(4d) 爷爷/去/乡下/**之前**/去了/城里，因为/那里/空气/清新/如洗。 Yeye/qu/xiangxia/**zhiqian**/qule/chengli/,/yinwei/nali/kongqi/qingxin/ruxi. Grandpa/go-to/countryside/**before**/went/city, because/there/air/fresh/as washing. *Before going to the countryside, Grandpa went to the city because the air there was fresh and pure.*

**Table 3 brainsci-12-00474-t003:** Means and SD (standard deviations) of the experimental items.

	Cloze Probability		Acceptability	
	Mean	SD	Mean	SD
*after-congruent*	0.487	0.26	5.35	0.71
*after-incongruent*	0.131	0.19	2.28	0.74
*before-congruent*	0.497	0.27	5.13	0.83
*before-incongruent*	0.123	0.01	2.27	0.73

**Table 4 brainsci-12-00474-t004:** Mean percentage of the closer referents chosen for the locative pronoun *nali* for each condition in forced choice task.

	Truncated Context		Whole Context	
	Mean	SD	Mean	SD
*after-congruent*	85.7	0.18	87.0	0.09
*after-incongruent*	83.2	0.17	36.2	0.14
*before-congruent*	90.5	0.12	97.4	0.10
*before-incongruent*	81.7	0.15	34.9	0.14

## Data Availability

The datasets presented in this article will be provided on request. Requests to access the datasets should be directed to the corresponding author.

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
