# Peer review of "Understanding Temporal Relations in Mandarin Chinese: An ERP Investigation"

_brainsci, 2022, doi:10.3390/brainsci12040474_

Round 1

Reviewer 1 Report

This is a valuable article that contributes to answering the question of how temporal connectives are processed in tenseless languages. 

Some small remarks.

  1. Many times the authors remark to English as an example of a tense language. Why not refer to tense languages that are more related like Korean and Japanese? As future research it may even be useful to conduct a comparative study of Mandarin, Korean and/or Japanese.
  2. Statistics: the authors use classical ANOVA tests. That is not wrong, but given the fact that we have a subject x item design here, it may be even better to use a linear mixed-effects regression analysis, where subjects and sentences are the random factors (see Chapter 7 of Harald Baayens's book at http://www.sfs.uni-tuebingen.de/~hbaayen/publications/baayenCUPstats.pdf . Or even a GAM may be useful, see slide 37 and more of Harald Baayen's presentation at https://lstat.kuleuven.be/research/lsd/lsd2012/presentations2012/presentation.pdf .
  3. Multiple times the author find a significant interaction. It may be useful to show interaction plots (for some of them) since they immediately provide insight into the nature of the interaction.
  4. Line 267: Replace 'As shown in Table 4 statistical analysis ...' by: 'Mean scores and standard deviations are given in Table 3. Statistical analysis ...'
  5. Refer in line 279 to Table 4.

Author Response

Please see the attached PDF responses

Reviewer 2 Report

This manuscript presents an ERP experiment that tests for processing correlates of temporal iconicity in the ordering of linguistic expressions in Mandarin. Prior work has indicated processing difficulties when the linear order of expressions does not match the temporal ordering of the denoted events (e.g. "before entering the house, she opened the door" where "opened the door" is the first event but second proposition in the utterance.) The authors reason that such effects may be sensitive to the presence or absence of overt tense morphology; prior languagess studied include such morphology, while Mandarin does not. They report a P600 ERP effect that is sensitive to the iconicity of how propositions are orders, as previouslsy observed in other languages, which is also sensitive to whether the temporoal expression is "before" or "after", and further correlates with working memory capacity of participants.   Overall I have a positive impression of this manuscript. The core experimental goal is well motivated theoretically and the design and analysis appear to build and connect with existing literature on this topic. I should note that I am not an expert in Chinese linguistics, nor am I an expert in the state-of-the-art in terms of temporal ordering effects, so my comments will focus on issues more broadly apparent as a psycholinguist with a background in sentence processing. Three issues are especially important for making this manuscript more impactful. First, there is ambiguity in the presentation in terms of the role of iconicity in expressions, compared to the specific temporal modifiers "before" and "after". Second, key aspects of the statistical analysis are not explicitly motivated and may lead to confusion. Third, the motivation for an individual difference analysis was not clearly laid out. I expand on these and some more minor points below.   Major points   I found myself getting confused at various parts about whether the key predictions concern the match between proposition order and event order, or whether certain key predictions were about the specific lexical items "before" (zhiqian) and "after" (zhihou) (e.g. lines 220-222). I *think* that the main point is actually abouto competing theoretical accounts: An iconicity account predicts that (mis)match will drive ERP responsses (regardless of lexical item), while a polarity account predicts that the lexical item used will matter. Of course, temporal adverb, iconicity, and plausibility are fully separable (English examples here):   1. Mary knocked before entering the houose (before, ICONIC, plausible) 2. Before entering the house, Mary knocked (before, NON ICONIC, plausible) 3. Before knocking, Mary entered the houses (before, ICONIC, not plausible) 4. Mary entered the houses before knocking (before, NON ICONIC, not plausible) 5. Mary knocked after entering the houose (after, ICONIC, implausible) 6. After entering the house, Mary knocked (after, NON ICONIC, implausible) 7. After knocking, Mary entered the houses (after, ICONIC, plausible) 8. Mary entered the house after knocking (after, NON ICONIC, plausible)   At the very least, these contrasting predictions were difficult at times to tease out. This comes up e.g. around line 83 ("support the idea that before is semantically/pragmatically more costly to understand than after[...] also differ in polarity") and again around lines 470-476 ("after iss consistent with such a [iconicity] preference[... but the] before construction not only reverses the linear order but also[...]"). In both of those passages, it seems that the lexical item is being confounded with whether the expression is or is not iconic. I think this issue an be addressed by clarifying the writing. But, if I'm misunderstanding more broadly the theoretical goals of the paper, then this is a more significant issue with how the experiment is presented.   Key aspects of the statistical analyses can be better motivated. This includes the choice to separately analyze midline electrodes as compared to lateral sites. Here, I would urge the authors to conduct a single analysis, perhaps with a 3 (left, midline, right) by 3 (anterior, middle, posterior) design for topography.   Another key question concerns how time-windows were selected: 300-800 ms is *not* a standard window for P600 comparisons. Ln 348 appears to indicate that time-windows were selected based on visual inspection fo the data, which is not appropriate. If an exploratory analysis is conducted across time-windows, then this needs to be stated very explicitly and may motivate adjustments to statistical thresholds. A related question arose around the discussion of individual differences correlations; Fig. 3 points to a 3000-4500 ms time-window along-side the 300-800 ms windows for target and sentence-final words; what is the motivation for the first of these time windows?   The individual differences analysis raised some questions -- some perhaps easy to address. I did not see a motivation for this analysis in the Introduction, but the Discussion makes clear that iconicity effects may be modulated by working memory capacity. This makes sense to me and I would just like to see the logic made clear in the Introduction. The use of the AQ and discussion of imagination made less sense to me and appeared to open up the degrees of freedom in in this analysis in terms of finding some relationship or another. This aspect of the analysis should either be explicitly theoretically grounded, or perhaps set aside.   Minor points   Throughout: I think "tenseless" is used to mean "lacks overt inflectional marking for tense". But, this terminology may be (mis?) interpreted to mean the language lacks tense altogether in terms of syntax-semantics mapping. If the former meaning is meant, this should be made explicit.   ln. 97: source for the frequency difference mentioned here?   Table 1: "veridicality" is never defined as it relates to this table   Figs 1/2:   - These figures are very low resolution; please ensure high quality - Plotting four lines on top of each other is very difficult to visually follow. I would suggest separating these plots into pairwise comparisons - Please add error intervals for the ERP traces following the "best practices" recommendations from Pernet et al. (2020 Nat Neurosci) - Indicate the time window for the topography plots

Author Response

Please see the attached PDF responses

Reviewer 3 Report

The manuscript entitled "Understanding temporal relations in Mandarin Chinese: An ERP investigation" described some interesting EPR findings on the neural mechanisms of understanding the temporal relations in the context of world knowledge. Overall, I applaud the authors for the excellent job of summarizing the literature, their own studies and results, and the clear discussion. I think the research will contribute to the research in semantic processing and be of interest to a general audience.

I have only one major suggestion. It is not entirely clear how WM fits in the study. The literature review focused mostly on the different theories and their predictions, but how WM relates to language comprehension is not well discussed in the intro. Given that there are some major interesting findings, I would suggest the authors have a section about WM and language comprehension in the intro. Also, it feels like coming from nowhere about the AQ. Why was it included? Any theoretical considerations?

A minor correction: 

Page 4, line 146, I think the label should be 3a instead of 3b.

Author Response

Please see the attached PDF responses

Round 2

Reviewer 2 Report

I am grateful for the authors' attention to my previous comments; this remains an interesting manuscript that will add value to the field. 

I do have one lingering concern and I leave to the editors to decide how to proceed. The authors' reply makes clear that, in Mandarin, iconicity is fully correlated with whether the connective is zhiqian (before) or zhihou (after). they remark, accordingly, that "the lexical difference can be well controlled when exploring the iconicity hypothesis." 

But, I disagree with this conclusion (or I misunderstand the authors' point). It seems that iconicity is *confounded* with the lexical choice of zhiqian or zhihou; if those terms differ in polarity, then iconicity is also confound with polarity. I think it would be helpful to clarify for the non-Mandarin-speaking whether these stimuli allow distinct theories to be teased apart.